# First Record of Microplastic Contamination in the Non-Native Dark False Mussel *Mytilopsis leucophaeata* (Bivalvia: Dreissenidae) in a Coastal Urban Lagoon

**DOI:** 10.3390/ijerph21010044

**Published:** 2023-12-27

**Authors:** Raquel A. F. Neves, Tâmara B. Guimarães, Luciano N. Santos

**Affiliations:** 1Graduate Program in Neotropical Biodiversity (PPGBIO), Institute of Biosciences (IBIO), Federal University of the State of Rio de Janeiro (UNIRIO), Avenida Pasteur 458, Rio de Janeiro 22290-240, Brazil; tamara.guimaraes@edu.unirio.br (T.B.G.); luciano.santos@unirio.br (L.N.S.); 2Research Group of Experimental and Applied Ecology, Department of Ecology and Marine Resources, Institute of Biosciences (IBIO), Federal University of the State of Rio de Janeiro (UNIRIO), Avenida Pasteur 458 Lab 307, Rio de Janeiro 22290-240, Brazil; 3Laboratory of Theoretical and Applied Ichthyology, Department of Ecology and Marine Resources, Institute of Biosciences (IBIO), Federal University of the State of Rio de Janeiro (UNIRIO), Avenida Pasteur 458 Lab 314A, Rio de Janeiro 22290-240, Brazil

**Keywords:** aquatic pollution, bivalves, brackish system, invasive species, plastic pollution

## Abstract

Microplastic contamination is a global concern due to its conspicuous presence in aquatic ecosystems and its toxic nature to environmental and human health. False mussels are among the most notable fresh- and brackish water invaders. The invasive *Mytilopsis leucophaeata* in Rodrigo de Freitas Lagoon-RFL (Rio de Janeiro, Brazil) is the most abundant macrofaunal invertebrate, widely established and distributed throughout the lagoon. This study aimed to assess microplastic contamination in this invasive filter feeder and evaluate its potential use as a bioindicator. Agglomerates (~100 mussels) were manually collected using a stainless-steel spatula in ten sampling areas distributed throughout the whole lagoon and kept frozen. In the laboratory, 60 individuals were sorted by area for soft-tissue digestion. Each pool of 10 soft-tissue mussels (*n* = 6 by area) was wet-weighted and then placed in a 150-mL decontaminated glass beaker with 50 mL of 10% KOH. Samples were heated (40 °C) for 48 h, and digested samples were filtered in glass-fiber membranes. Microplastics were found in all samples of mussels (*n* = 60) from RFL; the particles were mostly lower than 100 µm with a mean concentration (±SD) of 35.96 ± 47.64 MPs g wet-weight^−1^. Microplastics were distinguished in seven shapes with different occurrences in samples (%): fiber (43.3%); fragment (34.3%); film (16.3%); sponge/foam (4.9%); pellet (0.57%), rope/filaments (0.17%); and undefined (0.4%). Thirteen colors of microplastics were found, but transparent (54.94%), black (10.77%), and white (9.36%) were the most common. *Mytilopsis leucophaeata* were useful to assess microplastic contamination in RFL and might be preferentially used in other invaded brackish systems instead of native and often threatened bivalves. Our results confirm the effective application of bivalves as an indicator of coastal microplastic pollution.

## 1. Introduction

Coastal marine ecosystems are frequently impacted by anthropogenic activities, including the intentional and unintentional introductions of non-native species [1]. The most conspicuous cause of biological invasion in coastal systems is the introduction of non-native species through the ballast water of ships in areas adjacent to ports and marinas [2,3]. Several consequences in coastal ecosystems could be attributed to bioinvasion, such as the establishment of new ecological interactions (e.g., competition, predation, parasitism) in invaded systems, loss of native biodiversity, changes in the structure and functioning of the biological community, and potentially, alterations in physical and chemical properties of the system [4,5,6,7,8]. 

Marine and estuarine invertebrates have the highest absolute contribution, in terms of species numbers, to the total richness of non-native species in Brazilian coastal systems [9]. In the last decades, new records of the non-native bivalve *Mytilopsis* (Dreissenidae) spread its geographical distribution in brackish systems worldwide [10], as well as in Brazilian systems. The dark false mussel *Mytilopsis leucophaeata* (Conrad, 1831) is native to the Gulf of Mexico and the United States of America but is currently recorded in Europe, Asia, North Africa, and Brazil [10]. In Brazil, *M. leucophaeata* was accidentally introduced in brackish systems, probably through adult individuals attached to ship hulls or larval dispersion in ballast water [11,12,13]. The first record of this non-native species was in 2004 in estuaries of the state of Pernambuco in the Northeast country’s region [14] and, after a decade, *M. leucophaeata* was found in Rodrigo de Freitas Lagoon, Rio de Janeiro city, at the Southeastern region of Brazil [12]. The dark false mussel *M. leucophaeata* became the most abundant epibenthic macrofauna in the Rodrigo de Freitas Lagoon, being widely distributed throughout its littoral zone [13,15], concurrently with the report of a reduced density of the native bivalve *Brachidontes darwinianus* (Linnaeus, 1798) in this system [16]. Negative impacts of non-native bivalves on native species are associated with higher rates of seston filtration and food ingestion of the invaders, competition for space with native fouling species, shifts in habitat structure and refuge areas for benthic species, and changes in prey availability for higher trophic levels, altering community structure and food webs [8,13,16,17,18]. Moreover, the colonization of artificial substrates by *Mytilopsis* sp. seems to mediate its invasion in brackish systems [10].

Bivalves are filter-feeding organisms that capture phytoplankton and suspended organic matter in the water column. Their filter-feeding habits, high abundance, life cycle characteristics (e.g., long-living, sessile), and the ability to bioaccumulate contaminants make the bivalves excellent bioindicators of water quality [19,20,21]. Among emerging aquatic contaminants, microplastics have become a global concern due to their conspicuous presence worldwide and toxic nature to environmental and human health. Microplastics can also release leachates and additive chemicals (e.g., bisphenols and phthalates) that induce toxicity to biota, alter aquatic ecosystem functioning, and harm human health [22,23,24]. Evidence of microplastic transfer through aquatic food webs raises public health issues regarding human consumption of contaminated seafood, although the assessment of adverse effects on human health is still limited and difficult to directly identify [25]. Several studies have proposed the use of bivalves to understand the degree of environmental pollution, accumulation, and toxicity of microplastics in aquatic systems [26,27,28,29]. In a global view, the mean values of microplastic concentration in bivalves ranged from 0.04 to 20 particles g^−1^ [30]. However, other studies reported higher average concentrations in wild and farmed bivalves, such as 39.2 to 138 MPs g^−1^ and 89.4 to 259.4 MPs g^−1^, respectively [31]. Microplastics uptake by bivalves occurs through water filtration, and the size of microplastics ingested by them seems to be related to the size distribution of natural food items [30]. Moreover, the body length of bivalves also seems to influence the size of microplastics ingested [32]. Among studies that reported the size range of microplastics accumulated in bivalves worldwide, the proportion of small particles (<1000 µm) was significantly higher than large microplastics (1000–5000 µm) [30]. 

Despite its reduced size (~20 mm of total shell length), a single individual of *M. leucophaeata* can filter close to 55 mL of water h^−1^, approximately 480 L year^−1^ [33], consequently capturing countless suspended particles for its ingestion. Concerning its preferred range of ingested particles, the mean size of particles ingested by the bivalve *M. leucophaeata* varies from 15 to 40 μm, although it has been observed that individuals are able to capture microalgal cells smaller than 4 μm [11]. These characteristics make this non-native species an interesting candidate for investigating microplastic pollution; thus, the present study aimed to assess microplastic contamination in the non-native species *M. leucophaeata* from Rodrigo de Freitas Lagoon, which may highlight its potential use as an indicator of microplastic contamination. In addition, considering that dark false mussels colonize a wide variety of hard substrates, including artificial substrates made of plastic materials, this study tested differences in microplastic contamination in bivalves colonizing natural and artificial (i.e., man-made) substrates. Artificial substrates can affect the availability of nutrients and suspended particles in the water column of eutrophic systems by the removal of attached periphyton [34,35,36]. However, there is no information regarding its potential role as a source of microplastics for fouling biota. This study was developed to integrate a global effort of goals and initiatives for sustainable development (Life Below Water) to improve ocean health and environmental and human safety.

## 2. Materials and Methods

### 2.1. Study Area

The Rodrigo de Freitas lagoon is a coastal shallow system (mean depth of 2.8 m) located entirely within a densely urbanized area in the city of Rio de Janeiro (Southeastern Brazil). Its water surface covers approximately 2.2 km^2^, with a perimeter of 7.8 km and a volume of 6,200,000 m^3^. This coastal system receives inflows through three main freshwater systems: Macacos (7.9 km^2^ of hydrographic basin); Cabeça (1.9 km^2^); and Rainha (4.3 km^2^) rivers. During periods of intense rainfall, the lagoon receives polluted stormwater from these rivers, especially with organic waste, making it more susceptible to dissolved oxygen deficits [37]. In addition, the lagoon is frequently subjected to inputs of domestic sewage from surrounding neighborhoods, which is evidenced by the presence of enteric bacteria such as *Escherichia coli* [37]. To maintain the water level and quality, three floodgates artificially controlled by the municipal government are responsible for regulating the inflow of freshwater and the lagoon connection with the Atlantic Ocean, which sustains its brackish conditions. When opened, the floodgate located on General Garzon Street makes it possible for the rivers to flow into the lagoon, while the floodgate located on Visconde de Albuquerque Street allows those riverine waters to flow to the ocean. The exchange of water between the lagoon and the ocean is controlled by a floodgate located at Jardim de Alah Channel. However, this floodgate’s efficiency is frequently reduced due to sedimentation, requiring regular dredging to enable the exchange between the lagoon and the ocean [8]. Three different regions were delimited in the lagoon by the municipal regulation n° 18415/2000 for the implementation of monitoring programs. These regions were set considering the environmental conditions and water circulation in the lagoon, where region 1 (~70% of the total area) is mostly influenced by freshwater inflow from rivers and pluvial waters [37]. Regions 2 and 3 are more influenced by the exchange between the lagoon and ocean waters through the Jardim de Alah Channel [37]. Sediment in the lagoon is composed of very fine sand (Folk’s classification = muddy sand), and four phthalates (DBP, DnOP, DEHP, and DINP) were recently detected in this compartment of Rodrigo de Freitas Lagoon [38]. Nevertheless, Rodrigo de Freitas Lagoon is a popular tourist spot and holds significant landscape value, being surrounded by residences, commercial establishments, hospitals, equestrian and jockey clubs, and terrestrial and aquatic sports facilities. 

### 2.2. Dark False Mussel Sampling

The sampling of bivalves was carried out on 30 November 2021 throughout the whole perimeter and comprised three different regions of the lagoon, using a 15 hp motorized boat (Figure 1). Sampling was conducted following the approval of the Chico Mendes Institute for Biodiversity Conservation (ICMBio n° 81992-1) and the National System for the Management of Genetic Heritage and Associated Traditional Knowledge (SISGEN n° A1B005D). The individuals of *M. leucophaeata* were manually collected using a stainless-steel spatula by scraping the colonized substrate at each of the 10 sampling areas; at least 100 individuals were sampled per area. Hard substrata (e.g., natural or artificial) used for sampling varied among areas based on substrate availability in each sampling area and the occurrence of bivalve agglomerates. Rodrigo de Freitas Lagoon is an urban coastal system that provides both natural and artificial substrates for colonization; however, the distribution of these substrates is spatially segregated throughout the lagoon regarding its uses. Additionally, water temperature (°C) and salinity (ppt) were evaluated in sampling areas using a thermosalinometer (Hanna Instruments, HI98319). Biological samples were kept on ice until they were frozen (−20 °C) for subsequent laboratory procedures for microplastic extraction. 

### 2.3. Laboratory Procedures

Microplastic analysis was based on protocols already developed for bivalve mollusks; these methods are based on the digestion of bivalve soft tissues using potassium hydroxide (KOH) [28,39]. Previous results of interlaboratory comparisons evidenced that the KOH solution is an optimal method to digest mussel samples considering microplastic recovery, time, technical issues, and cost [40]. For this purpose, six replicates consisting of pools of ten individuals were evaluated per sampling site; thus, in total, the soft tissues of 60 individuals were used per area. The total shell length (mm) of collected individuals was measured using a digital caliper with 0.01 mm precision. Individuals with total shell lengths varying from 10 to 20 mm were sorted to be used in microplastic analysis to avoid an effect on individuals’ size. This size range comprises the most representative cohorts of *M. leucophaeata* in Rodrigo de Freitas Lagoon [15]. The individuals were previously removed from their shells using sterilized stainless-steel forceps, and their soft tissue was wet-weighted (w.w.) using a precision balance (0.0001 g) and then subjected to chemical digestion by adding 50 mL of 10% KOH solution. Samples were incubated in an oven at 40 °C for 48 h to completely digest bivalve soft tissues. Therefore, the digested solution was filtered through a glass-fiber membrane filter (Macherey-Nagel MN GF-5; 0.45 µm) using a filtration system and a vacuum pump. After filtration, the membranes were placed in previously washed and decontaminated glass Petri dishes and kept at room temperature in a desiccator with silica gel. Microplastics retained on the membranes were analyzed using a stereoscopic microscope with an integrated camera (Leica EZ4 HD). All microplastics found were quantified and characterized by type and color. To prevent cross-contamination of the samples, all materials used for sample collection and processing were washed and decontaminated according to specific protocols for microplastic analysis. In addition, blanks were performed in all analytical steps, and all the samples were kept covered as much as possible [41]. After microscope analysis, no airborne contamination was detected in blank samples (*n* = 10). 

### 2.4. Analytical Limitations 

One of the limitations of the present study was the imprecision of the stereoscopic microscope with a total magnification of 8–35× to measure microplastic particles lower than 100 µm. However, since most of the particles found in bivalves were lower than 100 µm, only the size range of microplastics was provided to avoid imprecisions. Another limitation was for the conduction of chemical analysis of microplastics found in bivalves using µ-FTIR (FTIR Nicolet 6700) in reflection mode. The difficulties faced in performing the µ-FTIR analysis were related to the following: (1) the reduced size of microplastic particles (<100 µm), which did not allow for their removal from the glass-fiber membrane to individually place them on gold plat support for µ-FTIR analysis in reflection mode; and (2) the glass-fiber membrane was not suitable as support to directly conduct µ-FTIR analysis. Only three particles of different shapes and colors could be perfectly removed from membranes and completely analyzed. Therefore, our methods to measure particle size and perform chemical analysis of microplastics must be improved for further studies.

### 2.5. Data Analysis

The mean values and standard deviation (±SD) were calculated for all the environmental and biological variables. Differences in the number of microplastics in dark false mussels (number g w.w.^−1^) by substrate type—artificial (*n* = 24) and natural (*n* = 36)—were tested using the nonparametric Mann–Whitney U test. The percentage of microplastic occurrence in bivalves by shape and color was determined for each analytical replicate based on the relative proportion (e.g., the number of MPs/total count) multiplied by 100. 

## 3. Results

### 3.1. Environmental and Biological Variables

Individuals of *M. leucophaeata* were associated with different natural and artificial substrates according to the sampling area. The most common natural substrate colonized by agglomerates of dark false mussels was mangrove branches (Table 1). A wide variation in structures was observed among artificial substrates, but most of them were composed of plastic material (Table 1). The mean values (±SD) of salinity and water temperature at the sampling areas were 13.0 ± 0.44 ppt and 29.59 ± 0.92 °C, respectively. There was no gradient in salinity and temperature in sampling areas throughout this coastal lagoon, which could potentially affect bivalve distribution or microplastic availability in the water column. Moreover, no difference in environmental data was detected among the three regions of the lagoon set by the municipality for monitoring purposes. Detailed information on salinity and water temperature by sampling area is available as Appendix A.

Among the 600 individuals of *M. leucophaeata* evaluated in the present study, the minimum and maximum shell lengths ranged from 11.02 and 25.92 mm, respectively. In terms of soft tissue, the maximum and minimum wet weight ranged from 0.0710 and 0.4790 g ind^−1^, respectively. On average (±SD), the shell length and biomass of *M. leucophaeata* bivalves from Rodrigo de Freitas Lagoon was 19.05 ± 0.10 mm and 0.2238 ± 0.0780 g w.w. ind^−1^, respectively. 

### 3.2. Microplastic Contamination in Bivalves

Microplastics were found in all the samples of *M. leucophaeata* from Rodrigo de Freitas Lagoon. In absolute numbers, the minimum and maximum number of microplastics per sample (i.e., pool of 10 individuals) were 6 and 216 particles, respectively, with a mean value of 57.06 ± 39.73 microplastics per sample. The concentration of microplastics (mean ± SD) found in bivalves was 35.96 ± 47.64 MPs g w.w.^−1^, while the minimum and maximum values of microplastic concentration in bivalves were 2.27 and 241.56 MPs g w.w.^−1^, respectively. No significant difference was found in microplastic contamination in bivalves colonizing natural and artificial substrates in Rodrigo de Freitas Lagoon (Mann–Whitney U test, *p* = 0.582; Figure 2). 

The microplastics found in the dark false mussel *M. leucophaeata* were classified into seven distinct shapes (Figure 3), with different percentages of occurrence (%): fiber (43.3%); fragment (34.3%); film (16.3%); sponge/foam (4.9%); pellet (0.57%); rope/filament (0.17%); and undefined particles (0.4%).

Regarding the colors of microplastics found in dark false mussels from Rodrigo de Freitas Lagoon, a total of 13 colors were identified: black; blue; white; transparent; red; green; multicolored; gray; orange; yellow; brown; pink; and purple. There was a higher percentage of occurrence of microplastics in the transparent color (54.94%), followed by black (10.77%) and white (9.36%) (Figure 4).

## 4. Discussion

Agglomerates of the dark false mussel *M. leucophaeata* were mostly detected in a natural hard substrate (mangrove branches) throughout the perimeter of Rodrigo de Freitas Lagoon. As part of an environmental revitalization project initiated over 30 years ago in Rodrigo de Freitas Lagoon, more than 4500 red and white mangrove trees have been planted on the lagoon’s shores to provide habitat and shelter for invertebrate and vertebrate species [42]. However, mangrove trees are not native to this coastal lagoon, which was primarily inhabited by plants characteristic of restinga biome (i.e., coastal sandy plain vegetation). In addition, the higher relative colonization of natural hard substrata was not expected in the present study since the colonization of artificial substrates was shown to favor the establishment and distribution of *Mytilopsis* bivalves in invaded systems [10]. *Mytilopsis* species can colonize different types of artificial substrates in invaded systems, such as human construction, metal materials, plastic materials, rope and mesh netting, vessels, and wood fragments [10]. In the present study, no significant difference was detected in microplastic contamination in bivalves colonizing natural and artificial substrates. Therefore, artificial substrates do not seem to act as potential sources of microplastics for *M. leucophaeata*, in spite of most of the artificial substrates being composed of plastic materials. It is important to highlight that precautions were taken during the sampling of bivalves, processing, and analysis of samples in the laboratory to avoid crossed contamination, and no airborne contamination was detected in analytical blanks. Field and laboratory precautions to avoid cross-contamination by microplastics are important strategies for improving quality assurance [41]. 

Since our study is the first record of microplastic contamination in *Mytilopsis* spp., it was not possible to discuss concentrations found in Rodrigo de Freitas Lagoon compared to other brackish systems where these bivalves occur. However, microplastic contamination was already detected in other freshwater dreissenids, such as the invasive zebra and quagga mussels (*Dreissena* spp.) [43,44]. The occurrence of microplastics in freshwater dreissenids was described as widespread and variable in a study performed at Lake Michigan and a wastewater treatment plant [43]. Higher concentrations of microplastics were detected in 10–15 mm individuals of *Dreissena* spp. deployed at a wastewater treatment plant (~60 microplastics g w.w.^−1^), but those were not significantly different from concentrations found in mussels collected on a reference river at Lake Michigan [43]. In addition, dreissenids of the same size class (10–15 mm) deployed at an urban river and collected in two other reference rivers at Lake Michigan showed similar concentrations varying from 20 to 30 microplastics w.w.^−1^ [43]. In the present study conducted in an urban coastal lagoon, the mean concentration of microplastics in bivalves was 35.96 ± 47.64 g w.w.^−1^. Moreover, agreeing with results obtained in the present study, fibers were the most abundant microplastic particles (<70%), followed by fragments, found in the freshwater *Dreissena* spp. from Lake Michigan [43] and 100% of microplastics found in individuals of *D. polymorpha* from lakes in Turkey [45]. However, the most common colors of microplastics found in *Dreissena* spp. were clear (34%), blue (28%), black (14%), gray (12%), and red (8%) [43]. Globally, most of the microplastic particles found in marine environments are fibers (~91%) [46]. In the face of high environmental availability, bivalves are more likely to encounter fibers than the other microplastic shapes in the field [46,47,48]. In experimental trials, the mussels *Mytilus edulis* and *Perna viridis* also showed higher uptake rates of fibers when simultaneously offered with beads and fragments [48]. 

In comparison to other bivalves found in coastal lagoons worldwide, the microplastic concentration found in *M. leucophaeata* from Rodrigo de Freitas Lagoon in Brazil was higher than in mussels and oyster species evaluated in previous studies (Table 2). Microplastics were omnipresent in all 60 samples of bivalves evaluated in the present study. This fact may be explained by the location of Rodrigo de Freitas Lagoon in the metropolitan area of Rio de Janeiro city, where it is surrounded by busy public roads that connect the different zones of the city. This easy-access urban lagoon is highly visited by the local population and tourists year-round for leisure and sports activities, being more exposed to incorrect waste disposal, especially plastics. This shallow coastal system is also continuously subjected to inputs of illegal sewage and pluvial waters that carry urban residuals into the bay [37]. In addition, the filter-feeding behavior of *M. leucophaeata* can filter huge amounts of water and quickly clear high concentrations of suspended particulate matter from the water column [16,33,49]. Therefore, when microplastics are available in the brackish water of coastal systems, it is expected that individuals of *M. leucophaeata* might efficiently uptake them through water filtration. It is important to highlight that bivalves contaminated with microplastics apparently removed from the water column may also affect the dynamics of microplastic availability from planktonic to benthic compartments through particle elimination from feces or pseudofeces [50] and predation of contaminated bivalves [50]. Moreover, as previously described for other aquatic organisms, ingested microplastics may suffer biofragmentation turning into smaller particles then eliminated in feces [51,52], which can make these particles available to smaller detritivorous in aquatic environments. Therefore, our findings evidence the microplastic contamination in Rodrigo de Freitas Lagoon and highlight a concern regarding seafood contamination in this lagoon. Historical exploitation of fisheries (e.g., common snook, mugil, tilapia, and blue crabs) has been carried out by fishermen from the artisanal association located at this lagoon. Therefore, further studies are needed to assess the risks to human health regarding the consumption of fishery resources from Rodrigo de Freitas Lagoon, which may be contaminated by microplastics. Other contaminants have already been detected in Rodrigo de Freitas Lagoon, such as metals [53,54] and phthalates (DBP, DnOP, DEHP, and DINP) [38], enhancing the risks for human safety by direct environmental exposure or through the food chain. 

This study is the first to evidence microplastic contamination in the dark false mussel *M. leucophaeata*, as well as in a bivalve species of the genus *Mytilopsis*. This genus is known as a brackish water equivalent of zebra mussels [11], already registered in 108 introduced systems [10]. Among the extant species of *Mytilopsis* (*n* = 5), *M. leucophaeata* and *M. sallei* are the most widespread, with 88 and 46 occurrence records, respectively, in native and non-native areas [10]. Therefore, considering the widespread distribution of *Mytilopsis* species, further studies assessing microplastic contamination in brackish dreissenids are needed to evaluate its potential use as a sentinel species of microplastic pollution in coastal systems worldwide. 

Non-native dreissenids may negatively impact local biodiversity, leading to reductions in populations of native species and changes in ecosystem structure and functioning [62,63,64,65,66,67,68]. Fluctuations in the population of the native bivalve *Brachidontes darwinianus* in Rodrigo de Freitas Lagoon have been recorded after the dark false mussel’s introduction in 2014 [12,13,16]. Considering our results, *M. leucophaeata* may be preferentially used as a bioindicator or sentinel species of microplastic contamination in Rodrigo de Freitas Lagoon, as well as in other invaded systems, instead of native and often threatened bivalves. Environmental applications of invasive bivalves are being suggested for the recovery of eutrophic systems, water disinfection, and removal of organic and metal contaminants from invaded systems [69,70]. However, pest management and control must be combined with that environmental approach to avoid further spreading of non-native bivalves into new systems. 

## 5. Conclusions

This first record of microplastic contamination in the non-native brackish bivalve *Mytilopsis leucophaeata* highlights high concentrations ($\bar{X}$ = 35.96 g w.w.^−1^) of low-size particles (<100 µm) found in dark false mussels from the coastal urban Rodrigo de Freitas Lagoon (Rio de Janeiro state, Brazil). This study is also the first evidence of microplastic contamination in this coastal lagoon historically used for fishing activity and aquatic sports. Microplastic contamination in these bivalves was not significantly affected by the substrate of colonization (i.e., natural or artificial hard substrate). Most of the microplastics were characterized as fiber, followed by fragments and film. Thirteen different colors of microplastic were found in dark false mussels, but transparent, black, and white were the most found. *Mytilopsis leucophaeata* may be preferentially used to monitor microplastic contamination in Rodrigo de Freitas Lagoon and other invaded brackish systems instead of native and often threatened bivalves. 

## Figures and Tables

**Figure 1 ijerph-21-00044-f001:**
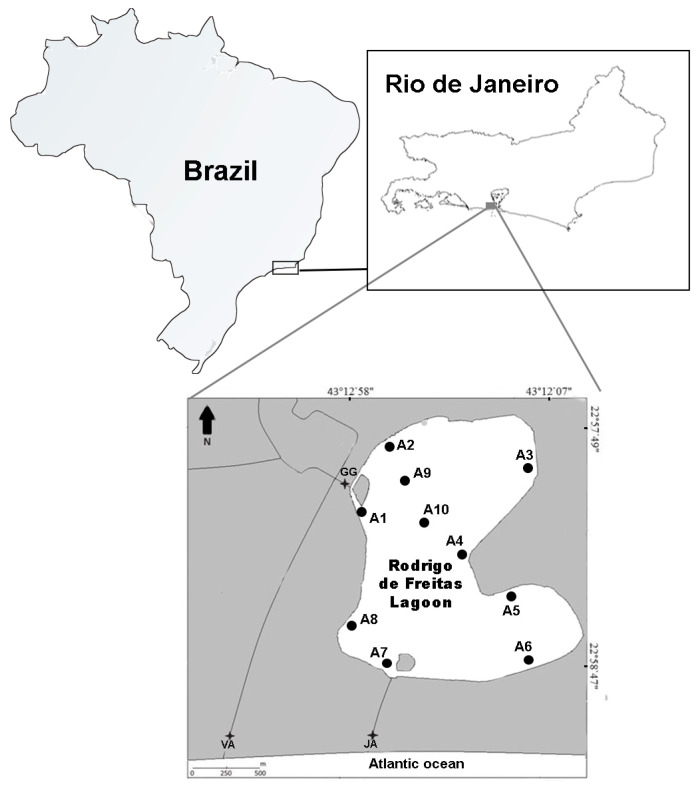
Geographical location of Rodrigo de Freitas Lagoon in Rio de Janeiro city, Southeastern Brazil. Sampling areas are indicated by • and their respective code in the map (A1–A10). An aerial view of the lagoon was provided in the Appendix A.

**Figure 2 ijerph-21-00044-f002:**
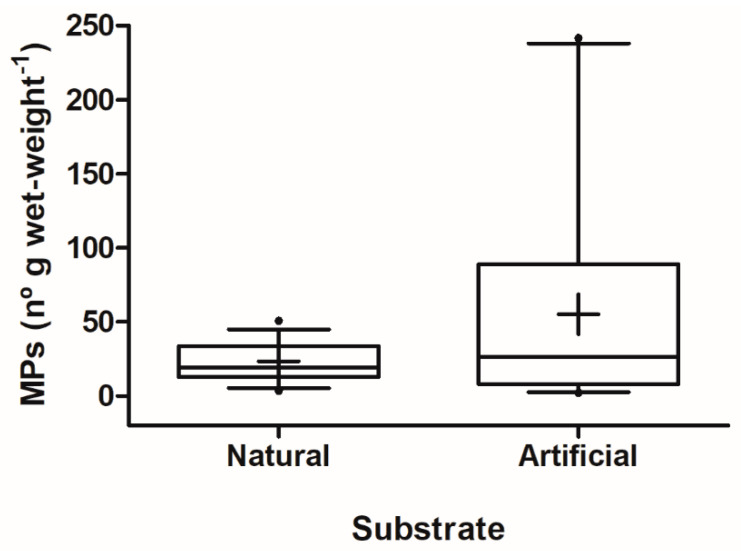
Box-and-whiskers plots for microplastic contamination in the dark false mussel *Mytilopsis leucophaeata* colonizing natural (*n* = 36) and artificial (*n* = 24) substrate. Median (full line inside the box), mean (+), outliers (•), standard deviation, and whiskers (5–95 percentile) are shown in plots.

**Figure 3 ijerph-21-00044-f003:**
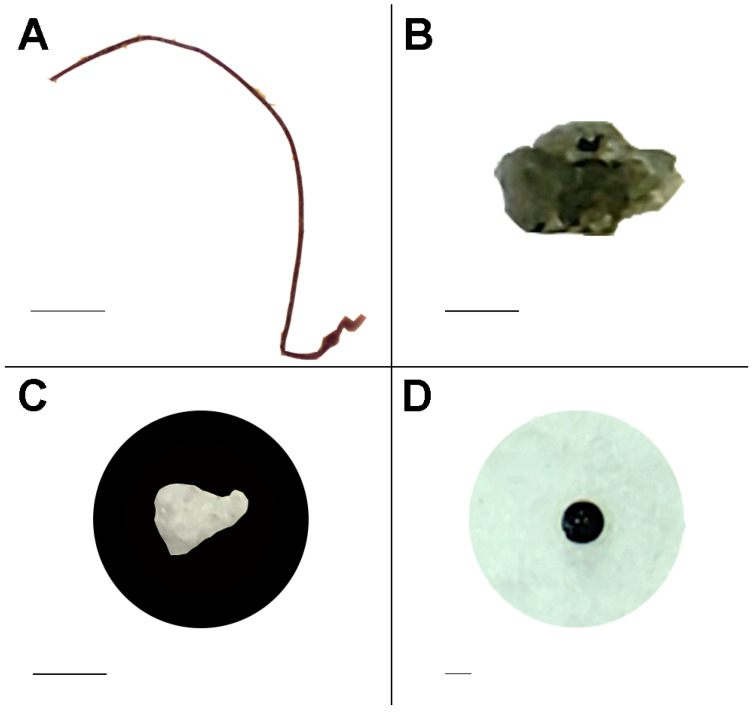
Examples of distinct shapes of microplastics found in the invasive bivalve *Mytilopsis leucophaeata* in Rodrigo de Freitas Lagoon. (**A**) fiber, (**B**) fragment, (**C**) styrofoam, and (**D**) pellet. Scale bars: 500 µm (**A**); 200 µm ((**B**) and (**C**)); and 100 µm (**D**).

**Figure 4 ijerph-21-00044-f004:**
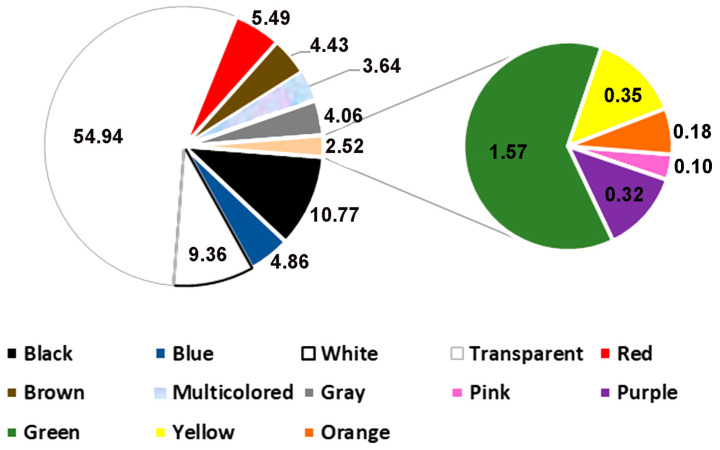
Percentage of microplastic occurrence by color (%) in soft tissues of individuals of *Mytilopsis leucophaeata* from Rodrigo de Freitas Lagoon.

**Table 1 ijerph-21-00044-t001:** Substrates used for *Mytilopsis leucophaeata* colonization in the ten sampling areas, and total number of individuals sampled by area. Sampling areas were distributed throughout the whole lagoon, and they were located within the three regions set by municipality regarding environmental conditions and water circulation in Rodrigo de Freitas Lagoon (Rio de Janeiro state, Brazil).

Sampling Area	Individuals	Substrate Type	Description
A1	100	Natural	Mangrove branches
A2	100	Natural	Mangrove branches
A3	100	Artificial	PVC pipe
A4	100	Natural	Mangrove branches and gravel
A5	100	Natural	Mangrove branches
A6	100	Natural	Gravel
A7	150	Artificial	Wooden pier
A8	100	Natural	Rocks
A9	100	Artificial	Floating buoys for sport activities
A10	100	Artificial	Buoy platform for water monitoring

**Table 2 ijerph-21-00044-t002:** Contamination of microplastics in bivalve species from coastal lagoons worldwide, showing the mean concentration of microplastics in bivalves (number of MPs g w.w.^−1^), the major class of particle size (mm), the percentage of dominant shape and color (i.e., ≥9%) of microplastics, and the geographical location of lagoons evaluated in previous studies.

Bivalve	Species	MPs g w.w.^−1^	Size (mm)	Shape (%)	Color (%)	Location	References
*Clam*	*Ruditapes decussatus*	1.40	0.1–1	Fiber 90%	Black 37.3% Blue 29.3% White 11.6%	Bizerte Lagoon, Tunisia	[55]
	*Polititapes* spp.	11.9	0.1–1	Fiber 82% Foam 16%	Blue 55% White 20% Violet 16%	Ria Formosa, Portugal,	[56]
	*R. decussatus*	18.4		Fiber 88% Foam 9.5%	Blue 58% White 18% Violet 10% Red 9%		
*Cockle*	*Cerastoderma* spp.	10.4	0.1–1	Fiber 94%	Blue 40% White 25% Violet 20%	Ria Formosa, Portugal	[56]
	*Cerastoderma eduli*	0.83–5.1	0.13–0.73	Fiber 37.5% Spherule 25% Fragment 25% Film 11.3%	Transparent 35.6% Yellow 16.9% Brown 10.6% Blue 10% Red 9.4%	Aveiro Lagoon, Portugal	[57]
*Mussel*	*Mytilopsis leucophaeata*	35.96	<0.1	Fiber 43% Fragment 34% Film 16%	Transparent 54.9% Black 10.7% White 9.4%	Rodrigo de Freitas Lagoon, Brazil	Present study
	*Mytilus* spp.	1.2	0.1–0.25	Fiber 50% Film 22% Spherule 18% Fragment 10%	Transparent 47% Brown 15% Orange 11%	Aveiro Lagoon and Formosa Lagoon, Portugal	[58]
	*Mytillus galloprovincialis*	0.80	0.1–1	Fiber 95%	Green 41% Red 24% Black 15% Transparent 10% Blue 10%	Bizerte Lagoon, Tunisia	[55]
	*M. galloprovincialis*	0.73	>0.025	Fragment 60% Line 20% Pellet 20%	Blue 69% Brown 16%	Ria Formosa Lagoon, Portugal	[59]
	*M. galloprovincialis*	0.77–4.3	0.16–1.4	Fiber 40% Film 27% Fragment 20%	Transparent 42.5% Orange 25% Black 15% Red 10%	Aveiro Lagoon, Portugal	[57]
	*M. galloprovincialis*	2.1	n.a.	* Fiber > Fragment > Film > Pellet	Clear colored 61.6% Black 10.3% Blue 10.3%	Bizerte Lagoon, Tunisia	[60]
*Oyster*	*Crassostrea gigas*	1.40	0.1–1	Fiber 90% Fragment 9%	Blue 37% Red 36%. Black 18.3%	Bizerte Lagoon, Tunisia	[55]
	*C. gigas*	3.40	n.a.	Fiber 80% Fragment 20%	Transparent 46.7% Blue 19.7% Green 14% Black 12%	Laguna de Terminos, Mexico	[61]

n.a. not available; * percentage values were not available.

## Data Availability

Data will be made available on request.

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
