# Peer review of "First Record of Microplastic Contamination in the Non-Native Dark False Mussel Mytilopsis leucophaeata (Bivalvia: Dreissenidae) in a Coastal Urban Lagoon"

_ijerph, 2023, doi:10.3390/ijerph21010044_

Round 1

Reviewer 1 Report

Comments and Suggestions for Authors

Authors present the microplastic ingestion in potentially pollution bioindicator species of bivalve in Rodrigo de Freitas Lagoon. The overall quality of the manuscript is good for a preliminary study. However I would suggest some improvement for this manuscript:

1. the picture of the lagoon of should be included in the Supplementary. (line 105-115)

2. Author used KOH to digest the tissue, where there a lot of concern using KOH that will diminish the color of microplastics. Did author test this?  (line 144-145, 153-154)

3. Provide the name of Glass fiber membrane 0.45 um pores (Whatman GF/C?) (Line 156)

4. What is the spatial resolution for stereoscopic microscope with an attached camera (Leica) used here? (Line 159-160)

5. Did author measure the size/length of Microplastics? (Line 160-161)

6. Did author find any airborne microplastics in the procedural blank used? (Line 163-165)

7. Figure 3- missing captions. and The pictures of Microplastics are not really clear/good resolution.

8. Author did not analyze the polymer- how author confirm all the particles found were indeed Microplastics? this is a very important question to be answered...

9.Assumption of plastics contaminants/additives/co-pollutants that may also present in the bivalve (as describes in Line 284-288) is very strong and heavy assumption/speculation, especially when author did not actually test the polymer of microplastics. Not all polymers will contain bisphenol or phtalates. Is there any other study in similar site/ similar species that reported the type of polymer?

10. Is there any selectivity in feeding mechanism of this invasive species? color selectivity? of shape?

11. How will size of microplastic influence the feeding?

Author Response

Response to Reviewers

Reviewer #1 

Authors present the microplastic ingestion in potentially pollution bioindicator species of bivalve in Rodrigo de Freitas Lagoon. The overall quality of the manuscript is good for a preliminary study. However I would suggest some improvement for this manuscript:

  1. the picture of the lagoon of should be included in the Supplementary. (line 105-115)

Authors: A picture of Rodrigo de Freitas Lagoon was included in the Supplementary Material (Fig. S1).

  1. Author used KOH to digest the tissue, where there a lot of concern using KOH that will diminish the color of microplastics. Did author test this?  (line 144-145, 153-154)

Authors: KOH is widely used by different authors since it’s effective for microplastic extraction from mussel samples (e.g., Li et al., 2019; Phuong et al., 2018). Based on the results of a previous study that performed an interlaboratory comparison of microplastic extraction methods from marine biota tissues, KOH solution was considered optimal to digest fish and mussel samples regarding microplastic recovery, time, technical issues, and cost (Tsangaris et al., 2021). Therefore, as previous studies already tested the methods for microplastic extraction from mussels, it was not necessary to test the efficiency or limitations of KOH solution. We have included a sentence in the revised version of the manuscript to justify the use of KOH as extraction method (Lines 168-170).

References

Li, J., Lusher, A.L., Rotchell, J.M., Deudero, S., Turra, A., Bråte, I.L.N., Sun, C., Shahadat Hossain, M., Li, Q., Kolandhasamy, P., Shi, H., 2019. Using mussel as a global bioindicator of coastal microplastic pollution. Environ. Pollut. 244, 522–533. https://doi.org/10.1016/j.envpol.2018.10.032

Phuong, N.N., Zalouk-Vergnoux, A., Kamari, A., Mouneyrac, C., Amiard, F., Poirier, L., Lagarde, F., 2018. Quantification and characterization of microplastics in blue mussels (Mytilus edulis): protocol setup and preliminary data on the contamination of the French Atlantic coast. Environ. Sci. Pollut. Res. 25, 6135–6144. https://doi.org/10.1007/s11356-017-8862-3

Tsangaris, C., Panti, C., Compa, M., Pedà, C., Digka, N., Baini, M., D’Alessandro, M., Alomar, C., Patsiou, D., Giani, D., Romeo, T., Deudero, S., Fossi, M.C., 2021. Interlaboratory comparison of microplastic extraction methods from marine biota tissues: A harmonization exercise of the Plastic Busters MPAs project. Mar. Pollut. Bull. 164, 111992. https://doi.org/10.1016/j.marpolbul.2021.111992

  1. Provide the name of Glass fiber membrane 0.45 um pores (Whatman GF/C?) (Line 156)

Authors: It was used the glass-fiber membrane from Macherey-Nagel MN GF-5. This information was included in the revised version of the manuscript (Line 182).

  1. What is the spatial resolution for stereoscopic microscope with an attached camera (Leica) used here? (Line 159-160)

Authors: We are not sure about the reviewer question, so we added here more information about the stereoscopic microscope used in the study. The Leica EZ4 HD has fixed eyepieces (10 x/ 20 mm) and provides a zoom range of 0.8 – 3.5 x (i.e., total magnification of 8 – 35 x). The microscope has an integrated camera that capture the images directly to the PC. We included the microscope model in the revised version of the manuscript (Line 186).

More information about the equipment can be found in the link below:

https://microscopecentral.com/products/leica-ez4-hd-digital-stereo-microscope

  1. Did author measure the size/length of Microplastics? (Line 160-161)

Authors: One of the study limitations was the imprecision of the stereoscopic microscope Leica EZ4 HD with a total magnification of 8 - 35 x to measure the microplastic particles lower than 100 µm. Most of the particles found in samples were lower than 100 µm, thus precise size of microplastics were not evaluated in the present study to avoid imprecisions.

  1. Did author find any airborne microplastics in the procedural blank used? (Line 163-165)

Authors: No contamination of airbone microplastic was detected in blank samples. This information was included in the revised version of the manuscript (Lines 190-191).

  1. Figure 3- missing captions. and The pictures of Microplastics are not really clear/good resolution.

Authors: All the images were captured by the integrated camera of Leica EZ4 HD stereoscope microscope. Since the microplastic particles are very small (< 100 µm), sometime to increase the zoom in a target particle to better show its characteristics can decrease image resolution. Authors have modified the Figure 3 to improve its resolution. It was not clear what’s missing in the captions, authors included more information.

  1. Author did not analyze the polymer- how author confirm all the particles found were indeed Microplastics? this is a very important question to be answered...

Authors: For the chemical analysis of microplastics, µ-FTIR analysis (FTIR Nicolet 6700) was conducted in refection mode being necessary the use of a gold plat as support. However, we have faced some difficulties to conduct µ-FTIR analysis in reflection mode since (1) most of the microplastics found in mussel samples were smaller than 100 µm making difficult its removal from the glass fiber membrane for analysis, and (2) glass fiber membrane was not suitable as substrate to directly conduct the chemical analysis. Only three particles of different shapes and colors could be perfectly removed from the membranes and completely analyzed, each spectrum was automatically compared to a reference database. The spectra were recorded between 4000 and 600 cm−1 with a resolution of 4 cm−1, 64 scans and an optical velocity of 1.26. Considering our analytical limitations, we are working on improvements of the methods applied to permit a robust chemical detection in further studies. But, as a preliminary result just to answer the Reviewer #1, we have detected polypropylene (PP) and polystyrene (PS) polymers in the samples analyzed. For further studies, we are collecting new samples from different compartments (biota, sediment, and water) to test more methods for chemical analysis. We have included a section in Material and Methods (2.4 Analytical limitations) in the revised version of the manuscript with an explanation of our limitations to conduct chemical analysis (Lines 193-206).

9.Assumption of plastics contaminants/additives/co-pollutants that may also present in the bivalve (as describes in Line 284-288) is very strong and heavy assumption/speculation, especially when author did not actually test the polymer of microplastics. Not all polymers will contain bisphenol or phtalates. Is there any other study in similar site/ similar species that reported the type of polymer?

Authors: This study was the first analysis of microplastics in this coastal lagoon, there is no additional information about microplastic contamination in the biota or environment. In a previous study, we have chemically analyzed the sediment of this lagoon for bisphenols, phthalates and HPAs (Neves et al., 2023). Four phthalates (DBP, DnOP, DEHP, and DINP) were detected in the sediment of Rodrigo de Freitas Lagoon, which highlight an environmental and human health concern. We have changed the sentence to avoid strong assumptions in the revised version of the manuscript (Lines 355-358).

Reference

Neves, R.A.F., Miralha, A., Guimarães, T.B., Sorrentino, R., Marques Calderari, M.R.C., Santos, L.N., 2023. Phthalates contamination in the coastal and marine sediments of Rio de Janeiro, Brazil. Mar. Pollut. Bull. 190, 114819. https://doi.org/10.1016/j.marpolbul.2023.114819

  1. Is there any selectivity in feeding mechanism of this invasive species? color selectivity? of shape?

Authors: It is not possible to discuss about selectivity in feeding mechanism of this bivalve species since we have not conducted microplastic analysis in water samples. It is an interesting topic to be considered in further studies.

  1. How will size of microplastic influence the feeding?

Authors: Considering that bivalves capture microplastic particles through water filtration, the size of microplastics available in the water column is directly related to their potential ingestion by bivalves. The literature indicates that the size range of particles preferentially ingested by M. leucophaeata varies from 15 to 40 μm, although some bivalves were able to capture microalgal cells smaller than 4 μm (Verween et al., 2010). Therefore, microplastics much larger than this size range are not expected to be captured by this invasive bivalve. A sentence was included in the revised version of the manuscript (Lines 85-90).

Reference

Verween A.; Vincx M.; Degraer S. Mytilopsis Leucophaeata: The Brackish Water Equivalent of Dreissena polymorpha? A Review. In The Zebra Mussel in Europe; Velde, G. Van der, Sanjeevi Rajagopal, Abraham bij de Vaate, Eds.; Backhuys Publishers: Leiden, The Netherlands, 2010; pp. 29–44 ISBN 9783823615941.

Reviewer 2 Report

Comments and Suggestions for Authors

Review of: First Record of Microplastic Contamination in the Non-Native Dark False Mussel Mytilopsis leucophaeata (Bivalvia: Dreissenidae) in a Coastal Urban Lagoon

Recommendation: Major revision

General comments

The manuscript by Neves et al. presents the First Record of Microplastic Contamination in the Non-Native Dark False Mussel Mytilopsis leucophaeata (Bivalvia: Dreissenidae) in a Coastal Urban Lagoon.

Specific comments

Major

- The environmental and biological factors investigated in the study are not appropriately used. Their measurements appear to serve no function. Find a use for them.

- what are the precautions taken especially when retrieving mussels from plastic substrates and cleaning processes etc. they must be clearly presented.  

- Blanks findings are not reported. How many were used? - Give the total number of mussels gathered in natural and artificial colonization substrates. Record them in a three-column table style with sample sites, natural and artificial substrates, and mussel numbers.

- Does Table 1 imply that the authors found no mussels on artificial substrates in places where they gathered on natural substrates? Clearly elaborate with proper explanations.

- Why is there a great difference on the substrates between sampled sites?  

- The discussion is filled with the first study's viewpoint. It must be reduced.

- There are no persuasive reasons for greater amounts of microplastics in the samples examined. it must be strengthened.

- The particle sizes must be estimated, and the findings must be reported and compared to other research. This cannot be skipped since particle size data is more important for risk evaluation.

- The major part of the results is based on natural and artificial substrates with no importance to the study area i.e., sampling sites. The discussion should focus on both sample sites and substrates.

- There is no in-depth examination or discussion of why artificial substrates contain more microplastics than natural substrates.

- Table 2 must include the size ranges of the microplastics reported in these studies.

- Add a section on the limitations of the study.

Minor

L84 – 86: Can authors give an overview of microplastic size ranges ingested as well as the number of microplastics by bivalve mollusks in the literature?

L164 – 165: please refer to and cite: https://doi.org/10.1016/j.teac.2023.e00203

L196: remove n = 60

Use same terminology in the main text. For e.g., sponge/foam or sponge/polystyrene

Fig. 3c does not look Styrofoam. Replace with another image.

Fig. 3a fiber must be of at least more than 1 mm of longest dimension. Is it quite normal in the samples analyzed in the mollusk samples worldwide? 

Table 2: Why are the present study's foam, pellet, filament results not mentioned here?        

Author Response

Response to Reviewers

Reviewer #2 

Review of: First Record of Microplastic Contamination in the Non-Native Dark False Mussel Mytilopsis leucophaeata (Bivalvia: Dreissenidae) in a Coastal Urban Lagoon

 Recommendation: Major revision

 General comments

The manuscript by Neves et al. presents the First Record of Microplastic Contamination in the Non-Native Dark False Mussel Mytilopsis leucophaeata (Bivalvia: Dreissenidae) in a Coastal Urban Lagoon.

 Specific comments

Major

- The environmental and biological factors investigated in the study are not appropriately used. Their measurements appear to serve no function. Find a use for them.

 Authors: Biological data was applied to standardize size (in length) of individuals analyzed and to convert the microplastic concentration (n° MPs g/ wet-weight-1). All data provided could be used for the further comparisons by other authors. Concerning the environmental data, it’s important to highlight that no salinity or temperature gradient was detected in this coastal lagoon, which could affect mussel distribution or, especially for salinity, might affect microplastics availability in the water column. Therefore, authors kept the environmental and biological data in the revised version of the manuscript.

- what are the precautions taken especially when retrieving mussels from plastic substrates and cleaning processes etc. they must be clearly presented.  

Authors: The precautions to avoid crossed contamination were the same independent of the substrate type. Mussels were manually collected using a stainless-steel spatula, individuals were stored in glass Pyrex containers and freeze until laboratory procedures. In the laboratory, only bivalves with closed valves were considered for microplastic analysis and their shells were washed with filtered distilled water before measurements and desiccation. All the laboratory procedures were performed using stainless-steel or glass items previously decontaminated for microplastic analysis. We have included the detailed information in the revised manuscript (Lines 149 and 294-300).

- Blanks findings are not reported. How many were used? - Give the total number of mussels gathered in natural and artificial colonization substrates. Record them in a three-column table style with sample sites, natural and artificial substrates, and mussel numbers.

 Authors: All the blanks were analyzed and no airbone microplastic was found in blanks; we have performed 10 blanks in the total. This information was included in the revised version of the manuscript (Lines 190-191).

A suggested by Reviewer #2, authors have included the number of mussels collected by sampling site in the Table 1.

- Does Table 1 imply that the authors found no mussels on artificial substrates in places where they gathered on natural substrates? Clearly elaborate with proper explanations.

Authors: The hard substrata (e.g., natural or artificial) used for mussel sampling varied among areas based on its availability in each sampling area and the occurrence of bivalve agglomerates. The Rodrigo de Freitas Lagoon is a coastal urban system that provides both natural and artificial substrates, but the distribution of these substrates are spatially segregated throughout the lagoon regarding its uses. We have improved this information in the revised version of the manuscript (Lines 153-155).

- Why is there a great difference on the substrates between sampled sites?  

Authors: That’s related to the environmental availability of substrates colonized by the invasive species in the different sampling areas.

- The discussion is filled with the first study's viewpoint. It must be reduced.

Authors: As suggested by Reviewer #2, authors have modified the Discussion in the revised version of the manuscript.

- There are no persuasive reasons for greater amounts of microplastics in the samples examined. it must be strengthened.

Authors: We have included more potential explanations in the Discussion section of the revised manuscript (Lines 330-336).

- The particle sizes must be estimated, and the findings must be reported and compared to other research. This cannot be skipped since particle size data is more important for risk evaluation.

Authors: One of the limitations of the present study was the imprecision of the stereoscopic microscope with a total magnification of 8 - 35 x to measure microplastic particles lower than 100 µm. Especially because most of the particles found in samples were lower than 100 µm, thus size of microplastics were not evaluated in the present study to avoid imprecisions. However, we included the information regarding the range of particles analyzed in the present study.

- The major part of the results is based on natural and artificial substrates with no importance to the study area i.e., sampling sites. The discussion should focus on both sample sites and substrates.

Authors: More information about sampling areas were provided in the revised version of the manuscript (Lines 129-134). However, it is important to highlight that most of the differences among areas are related to the substrates used for bivalve colonization. No difference was detected among areas in the environmental data measured in lagoon, which is expected considering its reduced size (2.2 km² of area).

- There is no in-depth examination or discussion of why artificial substrates contain more microplastics than natural substrates.

Authors: Actually, as previously reported in the study, there was no significant statistical difference between microplastic concentrations in mussels colonizing artificial and natural substrates. It’s known that artificial substrates can affect the availability of nutrients and suspended particles in the water column of eutrophic systems by the removal of attached periphyton (e.g., Sansone et al., 1998; Salant, 2011; He et al., 2017). However, there was no information about microplastic dynamics in artificial substrates and the potential role of artificial substrates (including the plastic items) as source of microplastics for attached biota, such as the bivalve M. leucophaeata. Authors included a sentence in the revised version of the manuscript to better explain this point (Lines 103-106).

References

He, H., Luo, X., Jin, H., Gu, J., Jeppesen, E., Liu, Z., Li, K., 2017. Effects of Exposed Artificial Substrate on the Competition between Phytoplankton and Benthic Algae: Implications for Shallow Lake Restoration. Water 9, 24.

Salant, N.L., 2011. “Sticky business”: The influence of streambed periphyton on particle deposition and infiltration. Geomorphology 126, 350–363.

Sansone, U., Belli, M., Riccardi, M., Alonzi, A., Jeran, Z., Radojko, J., Smodis, B., Montanari, M., Cavolo, F., 1998. Adhesion of water-borne particulates on freshwater biota. Sci. Total Environ. 219, 21–28.

- Table 2 must include the size ranges of the microplastics reported in these studies.

Authors: The size ranges of the microplastics reported in the studies were added in he Table 2.

- Add a section on the limitations of the study.

Authors: As suggested by Reviewer #2, the topic “Analytical limitations” was included in the Material and Methods section of the revised manuscript (Line  193-206).

Minor

L84 – 86: Can authors give an overview of microplastic size ranges ingested as well as the number of microplastics by bivalve mollusks in the literature?

Authors:

L164 – 165: please refer to and cite: https://doi.org/10.1016/j.teac.2023.e00203

 Authors: As suggested by Reviewer #2, the reference was included in the revised version of the manuscript (Line 190).

L196: remove n = 60

Authors: As suggested by Reviewer #2, the “n=60” was removed from the text in the revised version of the manuscript (Line 244).

Use same terminology in the main text. For e.g., sponge/foam or sponge/polystyrene

Authors: the terminology was standardized in the text of the revised manuscript.

Fig. 3c does not look Styrofoam. Replace with another image.

Authors: The image of styrofoam was replaced in the Figure 3.

Fig. 3a fiber must be of at least more than 1 mm of longest dimension. Is it quite normal in the samples analyzed in the mollusk samples worldwide? 

Authors: In face of the high environmental availability, bivalves are more likely to encounter fibers than the other microplastic shapes in the field (Andrady, 2017; Barrows et al., 2018; Qu et al., 2018). In experimental trials, the mussels Mytilus edulis and Perna viridis also showed higher uptake rates of fibers when simultaneously offered with beads and fragments (Qu et al., 2018). Regarding the size of microfibers in marine environments, they are predominantly 0.1 - 1.5 mm (Barrows et al., 2018). More information was included in the revised version of the manuscript (Lines 320-325).

Reference cited:

Andrady, A.L., 2017. The plastic in microplastics: A review. Mar. Pollut. Bull. 119, 12–22.

Barrows, A.P.W., Cathey, S.E., Petersen, C.W., 2018. Marine environment microfiber contamination: Global patterns and the diversity of microparticle origins. Environ. Pollut. 237, 275–284.

Qu, X., Su, L., Li, H., Liang, M., Shi, H., 2018. Assessing the relationship between the abundance and properties of microplastics in water and in mussels. Sci. Total Environ. 621, 679–686.

Table 2: Why are the present study's foam, pellet, filament results not mentioned here?     

Authors: In the Table 2, authors just included the percentage of dominant shapes and colors (i.e., percentage ≥ 9%) of microplastics found in bivalve species from coastal lagoons worldwide. In the present study, the percentage of occurrence of foam represented, respectively, 4.9%, 0.57% and 0.17%. The percentage of occurrence of all the microplastic shapes found is cited in the Lines 271-273.

Reviewer 3 Report

Comments and Suggestions for Authors

The current manuscript is satisfactory to be published after minor revisions as follows 

1. In methodology portion, did you analyzed the FTIR analysis of microplastics?

2. Refrences updated and satisfactory. Please add some lines to control the microplastics contamination in discussion with concern refrences. 

Regards 

Reviewer 

Author Response

Response to Reviewers

Reviewer #3

The current manuscript is satisfactory to be published after minor revisions as follows: 

  1. In methodology portion, did you analyzed the FTIR analysis of microplastics?

Authors: For the chemical analysis of microplastics, µ-FTIR analysis (FTIR Nicolet 6700) was conducted in refection mode being necessary the use of a gold plat as support. However, we have faced some difficulties to conduct µ-FTIR analysis in reflection mode since (1) most of the microplastics found in mussel samples were smaller than 100 µm making difficult its removal from the glass fiber membrane for analysis, and (2) glass fiber membrane was not suitable as substrate to directly conduct the chemical analysis. Only three particles of different shapes and colors could be perfectly removed from the membranes and completely analyzed, each spectrum was automatically compared to a reference database. The spectra were recorded between 4000 and 600 cm−1 with a resolution of 4 cm−1, 64 scans and an optical velocity of 1.26. Considering our analytical limitations, we are working on improvements of the methods applied to permit a robust chemical detection in further studies. But, as a preliminary result just to answer the Reviewer #1, we have detected polypropylene (PP) and polystyrene (PS) polymers in the samples analyzed. For further studies, we are collecting new samples from different compartments (biota, sediment, and water) to test more methods for chemical analysis. We have included a section in Material and Methods (2.4 Analytical limitations) in the revised version of the manuscript with an explanation of our limitations to conduct chemical analysis (Lines 193-206).

  1. References updated and satisfactory. Please add some lines to control the microplastics contamination in discussion with concern references. 

Authors: As suggested by Reviewer #3, a sentence about precautions to avoid crossed contamination was included in the Discussion section of the revised manuscript (Lines 296-300).

Reviewer 4 Report

Comments and Suggestions for Authors

The article presents results of research that may be of interest to a wide range of readers. The material presented in the article is actual in connection with the problem of global microplastic pollution of aquatic ecosystems. The object of research and methods are well described. The results are well discussed and presented in figures and tablesWithout any fundamental comments, I would like to express my proposal regarding the design of Fig. 4. Why was sector 5.38 selected for more detailed clarificationIt would be better to highlight the sector of 1.57 and less.

I recommend to accept the article for publication

Author Response

Response to Reviewers

Reviewer #4 

The article presents results of research that may be of interest to a wide range of readers. The material presented in the article is actual in connection with the problem of global microplastic pollution of aquatic ecosystems. The object of research and methods are well described. The results are well discussed and presented in figures and tables. 

Without any fundamental comments, I would like to express my proposal regarding the design of Fig. 4. Why was sector 5.38 selected for more detailed clarification? It would be better to highlight the sector of 1.57 and less.

Authors: As suggested by the Reviewer #4, the Fig. 4 was modified in the revised version of the manuscript.

I recommend to accept the article for publication

Round 2

Reviewer 2 Report

Comments and Suggestions for Authors

There are no further comments to add. Thanks for considering and improving the manuscript according to the reviewer`s comments.